# Impact of Operational Parameters on Droplet Distribution Using an Unmanned Aerial Vehicle in a Papaya Orchard

**Luis Felipe Oliveira Ribeiro [1]**, **Edney Leandro da Vitória [1,2,\*]**, **Gilson Geraldo Soprani Júnior [2]**,
**Pengchao Chen [3]** and **Yubin Lan [3]**

1. Department of Agricultural and Biological Sciences (DCAB), Federal University of Espirito Santo (UFES), São Mateus CEP 29932-540, ES, Brazil; luis.f.ribeiro@edu.ufes.br
2. Postgraduate Program in Tropical Agriculture (PPGAT), Federal University of Espirito Santo, São Mateus CEP 29932-540, ES, Brazil; dusoprani@gmail.com
3. National Center for International Collaboration Research on Precision Agricultural Aviation Pesticides Spraying Technology, College of Electronic Engineering and Artificial Intelligence, South China Agricultural University, Guangzhou 510642, China; pengchao.chen@scau.edu.cn (P.C.); ylan@scau.edu.cn (Y.L.)
* Correspondence: edney.vitoria@ufes.br or vitoria.edney@gmail.com; Tel.: +55-27-99943-0907

**Abstract:** Papaya production and export is increasingly expanding in the world market due to the nutritional importance of the fruit. Phytosanitary issues, labor shortages, and unevenness in land-based costal and motorized applications compromise crops, the environment, and humankind. The purpose of this study was to evaluate the efficiency of droplet distribution using an unmanned aerial vehicle, with different application rates (12.0, 15.0, and 18.0 L ha$^{-1}$) and spray nozzles (XR110015 and MGA015) in the upper (UL), middle (ML), and lower (LL) layers, and on papaya fruit clusters (BF). Water-sensitive paper labels and artificial targets were used to assess the efficiency. Coverage, density, droplet distribution, and droplet diameter were influenced by the application rates in the following order: 18.0 > 15.0 > 12.0 L ha$^{-1}$, showing concentrated droplet distribution in the respective layers: UL > ML > LL > BF. The 18.0 L ha$^{-1}$ rate increased the variables examined, and the droplet coverage on the UL using the XR110015 nozzle was 6.56 times greater than that found on the LL and BF. The MGA015 nozzle presented better results in the LL and BF in all variables analyzed. The UAVs were efficient in applying to the papaya crop and further studies should be carried out in order to confirm the efficacy of plant protection products applied using this technology.

**Keywords:** remotely piloted aircraft; drop distribution; application technology; efficiency; *Carica papaya* L.

## 1. Introduction

Papaya (*Carica papaya* L.) is one of the most commercially significant fruits in tropical and subtropical regions of the world, mainly for its nutritional and therapeutic benefits in the fruits, stems, leaves, and roots. The main economic significance is in the fruits, whether sold fresh or processed, as a source of vitamins A, C, potassium, folate (B9), niacin (B3), thiamine (B1), riboflavin (B2), iron, calcium, and fiber [1].

Brazil is the second leading producer and third leading exporter of papaya in the world [2]. According to [3], in Brazil, papaya production in 2021 amounted to 1,256,703 tons on about 28,495 harvested hectares. In Brazil, the states of Espírito Santo and Bahia are the largest papaya producers, yielding 439,550 and 400,438 tons in 2021, respectively, corresponding to more than 60% of the total national production [3].

Papaya production is at risk, given that the reduction in cultivated areas is a result of phytosanitary problems [2] since the crop is susceptible to pest and disease attacks [4–8]. The main pests include the white mite (*Polyphagotarsonemus latus*), the spider mite (*Tetranychus urticae*), among other tetranychid mites, aphids *Aphis gossypii* and *Myzus persicae,* and mealybugs, such as *Aonidiella comperei* and *Coccus hesperidum*. These are extremely harmful, as they cause direct damage to the fruit as well as indirect damage by transmitting viruses and stains,

in addition to being difficult to control. The main diseases that affect the culture of papaya are *Phoma caricae-payae*, black spot (*Asperisporium caricae*), and others.

The standard application methods for applying pesticides in the papaya crop are land-based applications, such as manual or motorized knapsack sprayers and hydropneumatic sprayers. However, when this equipment is not properly calibrated and regulated, it can cause damage during application and to the environment, often due to the excess of liquid sprayed. In addition, the lack of skilled labor reduces the efficiency of the application and the desired control effectiveness, since the crown architecture, height, and insertion of papaya fruits is variable, requiring the search for and improvement of new technologies in the application of pesticides in papaya [9].

Agricultural operations related to pest and disease controls by unmanned aerial vehicles is an emerging operational technique in many countries, and especially in Brazil and Latin America as a whole, such technique has been progressing more intensively in the last six years. This technique, when compared with ground applications, has a number of advantages related to social, environmental, and economic sustainability. The application technology through unmanned aerial vehicles (UAVs) is based on the use of pre-programmed flight plans, controlled by an autonomous operator in a ground station, and the equipment consists of a rotor, tank, spraying system, control system, environmental sensor, and power system, among others [10]. Compared to conventional ground and air applications, UAVs stand out due to the following advantages: ultra-low volume of solution, water saving, no need for a landing strip, less risk of pilot contamination [11–13], and no limitations on terrain shape, field size, crop pattern, and turning space [14].

The spray droplets' distribution is one of the most relevant parameters to identify the application process efficiency by unmanned aerial vehicles. Therefore, in the spraying and application process, when the specific characteristics of calibration, regulation, and consequent UAV settings are not observed, in order for the droplets to efficiently reach the target, this leads to pesticide waste, and a consequent reduction of effectiveness in the preventive or immediate action of the applied pesticide. Besides this, there are the undesirable effects of drift and runoff losses, increasing the risk of environmental and human contamination. According to [15], factors such as spray nozzles, weather conditions, operational flight speed and height, application range and rate, and flight path, are important operational parameters to be considered in the quality of pulverization using UAVs. There are a few relevant research studies on the application of the spraying technique in fruit orchards, however the number of studies is limited when compared with the advances in research on crops planted in total area (wheat, oats, corn, rice, and others). In the specific case of papaya, certain specific factors that are not observed in other fruit trees may be relevant in the process of droplet deposition, i.e., the size of the leaves and the position of the fruits concentrated in clusters. Within the research carried out in fruit orchards, we highlight the following: citrus [16–18], pineapple crop [19], pear crop [20,21], peach crop [22,23], apple crop [24–26], guava crop [27], mango crop [28], and grape crop [29], and a significant interaction of operational parameters on the application efficiency and/or efficacy of tested crop protection products and foliar fertilizers was observed in all of them.

In vineyards [29], the use of a high speed (3.0 m s$^{-1}$) enabled increased droplet deposition on the canopy and reduced losses to non-target areas using conventional spray nozzles. Furthermore, the very low spray application (53.0 L ha$^{-1}$) proved to be insufficient to provide application efficiency. In a peach orchard [22], the flight speed of 2.0 m s$^{-1}$ provided greater uniformity, droplet deposition along the canopy, and less spray droplet loss to the ground. In citrus [16], using a height of 1.40 m, there was a 59.6% improvement in droplet density in the lower layer. When spraying pineapple plants [19], to maximize the application efficiency and effectiveness, the operating height should be <2.5 m and the wind speed should be <5.0 m s$^{-1}$. Further studies were conducted in other shrub crops, such as conilon coffee [30,31], arabica coffee [32], and almond [33]. Significant differences in operational parameters during applications were observed in the droplet spectrum and the desired control efficacy with UAVs use in these studies.

Therefore, there is a great potential for using UAVs in orchards, due to labor flexibility, lower operational costs, and lower environmental impacts; however, these potential assumptions should be effectively evaluated before pesticide applications in order to understand the dynamics needed to configure the UAV. In this regard, the motivation of this work was to seek answers to questions and theoretical and experimental gaps that still exist in relation to the spraying and application of pesticides by UAVs in the papaya crop. In summary, we can highlight four gaps that justify this study: (i) The spraying and application of pesticides by means of UAVs is not commonly applied in the papaya crop, because there are doubts regarding the efficiency of the deposition of the different targets on the papaya plant; in practice, there is a preference for tractor-driven equipment that are extremely inefficient. (ii) There are numerous studies related to the efficiency of spraying and application of pesticides via UAVs in full-area crops (rice, corn, wheat, soybeans, among others), however, the same studies are extremely rare in fruit orchards. (iii) A papaya plant at productive age may present a series of pests and diseases in different locations along the plant canopy, from the highest leaves to the fruit cluster at the bottom; therefore, although there are studies on the effects of different operational parameters (height and operational flight speed, application rate, flight plan), they are not applicable to the papaya crop and should be properly investigated, initially on the application of different application rates and spray nozzles. (iv) Papaya is one of Brazil's most exported fruits to the European Union, Asia, and the United States, reflecting its commercial importance; however, the operational parameters for spraying and aerial application by UAVs have never been investigated.

Considering the lack of information regarding the quality of spraying and droplet deposition in the papaya crop, we decided to test the following hypotheses: (a) Higher application rates result in more regular distributions of the sprayed droplet on all leaf layers of the papaya plant, as well as on the fruit clusters. (b) The choice of the spray nozzle model is determinant to increase the penetration capacity and distribution of the sprayed droplets. (c) The iteration between the application rate and the nozzle model defines the best configuration of the UAV in applications on the papaya plant.

The purpose of this study was to evaluate the effect of spray nozzles and application rates on droplet distribution along the canopy height and fruit clusters of papaya plants using an unmanned aerial vehicle.

## 2. Materials and Methods

### 2.1. Experimental Area and Crop Characteristics

The research was conducted on a commercial farm, located in the district of Barra Seca, municipality of Sooretama, State of Espírito Santo, Brazil (Figure 1). The geographic coordinates of the experimental site are 19°07′56″ S and 40°06′51″ W. According to the Köppen classification, the climate of the research region is hot and humid, with a dry season in autumn–winter and a rainy season in spring–summer, type Aw [34].

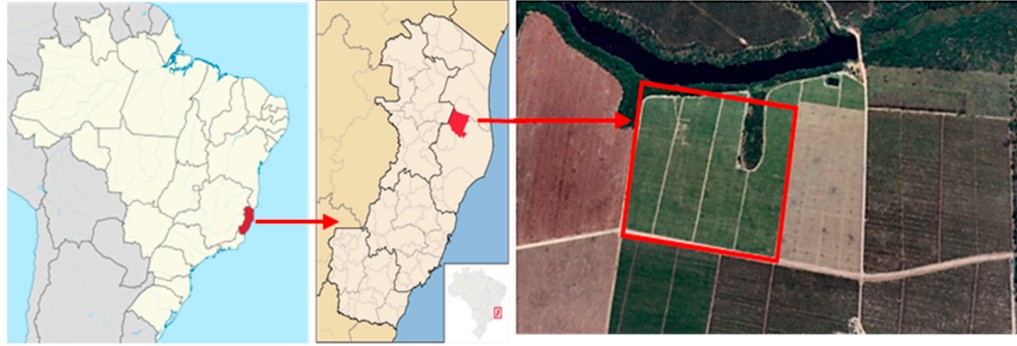

**Figure 1.** Location of the experimental area. The red frame indicates the location of the experimental treatments.

The field is a papaya plantation of the Tainung No. 1 variety, with planting spacing of 3.0 m × 2.0 m, totaling a stand of 1667 plants per hectare. At the time of the experiment, the plantation had an average maturity of 20 months, with an average height of 2.40 m.

### 2.2. Unmanned Aerial Vehicle Characterization

An unmanned aerial vehicle (UAV), brand DJI model T10 (Figure 2), was used, with a 10 L capacity, which was previously regulated and calibrated before application at the time of the experiment.

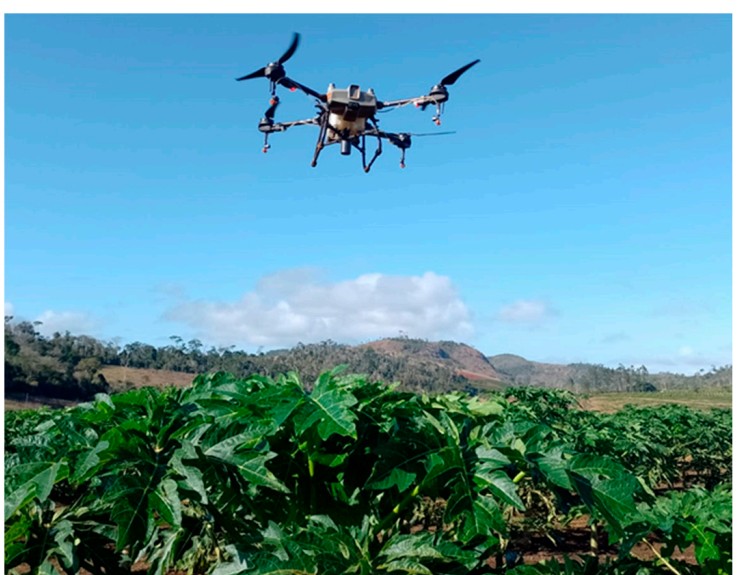

**Figure 2.** Multi-rotor unmanned aerial vehicle used in the spraying experiment.

Besides the tank for the product storage, the UAV was equipped with a water pump, a pipe circuit for liquid circulation, electronic control, valves, and other components. A flight speed of 5.0 m s$^{-1}$ and an average flight height of 2.5 m above the crown of the papaya plants were kept constant for all treatments. The main specifications are listed in Table 1.

**Table 1.** DJI T10 UAV specifications.

| | |
|---|---|
| Operating efficiency per hour | 15 acres |
| Number of rotors | 4 |
| Maximum operational flight speed | 0 to 7 m s$^{-1}$ |
| Maximum level flight speed | 4 to 10 m s$^{-1}$ (with strong GNSS signals) |
| Maximum bearable wind speed | 0 to 8 m s$^{-1}$ |
| Tank capacity | 10 L |
| Maximum effective spray width | 3 to 5.5 m |
| Stationary flight duration | 0 to 17 min |
| Maximum spraying flow | 1.81 L/min |
| Number of nozzles | 4 |

### 2.3. Experimental Design

The experiment was conducted in a randomized block design (DBC), with the treatments arranged in strips, with the first factor being the spray nozzles (flat-fan nozzle model XR110015 Teejet® and the hollow cone nozzle model MGA015 Magnojet®) and the second factor being the application rates (12.0, 15.0, and 18.0 L ha$^{-1}$). The treatment was repeated six times. Table 2 shows the experimental treatments' data.

**Table 2.** Experimental treatments.

| Treatments/Configuration | Spray Nozzle | Application Rate (L ha)$^{-1}$ | Pressure (kPa) | Flow Rate (L min)$^{-1}$ | Drop Classification * |
|---|---|---|---|---|---|
| T1 | | 12.0 | | | |
| T2 | XR110015 | 15.0 | | 0.59 | Fine drop |
| T3 | | 18.0 | | | |
| T4 | | 12.0 | 300 | | |
| T5 | MGA015 | 15.0 | | 0.57 | Very thin |
| T6 | | 18.0 | | | |

\* Droplet size classification corresponding to the nozzle manufacturers, XR and MGA, TeeJet® and MagnoJet®, respectively.

The area occupied by each experiment was 1050 m² (17.5 m × 60.0 m). Two central rows (useful row) of each treatment with six plants were used for the evaluations, considering one planting row as a border for each treatment, in order to avoid effects between treatments. The useful row was determined according to the effective strips of application, which were 2.5 m. Figure 3 shows the implemented experimental design.

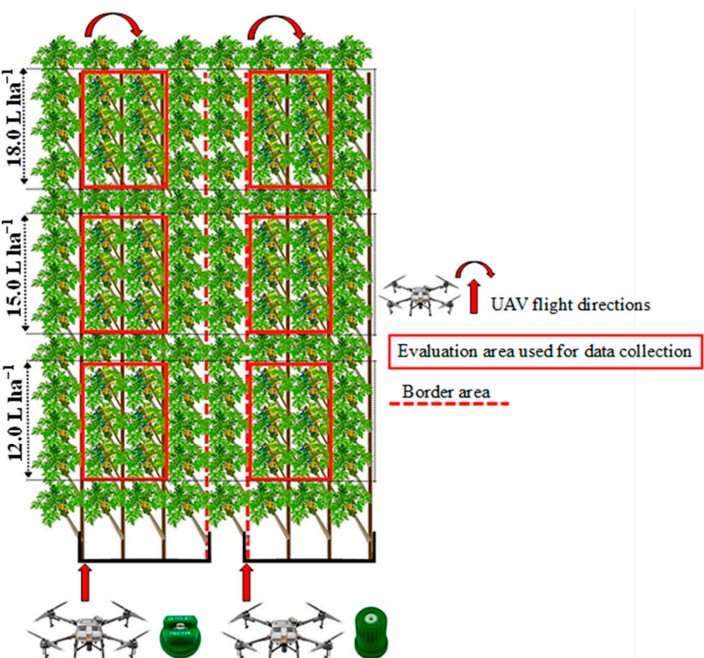

**Figure 3.** Experimental design of the experiment.

*2.4. Determination of Variables Related to Application Efficiency*

Water-sensitive paper labels with 76 mm × 26 mm dimensions were used in order to determine the sprayed droplet spectrum, as well as to estimate the application efficiency variables: coverage, droplet density, and the volumetric median diameter (VMD). In relation to the VMD, it is known that the values may vary due to the spread factor of the water-sensitive paper, however, considering that this interference is the same for all treatments, the VMD can be used as a comparative estimate of the size of the drops generated by each of the tips used in the treatments. In addition to the water-sensitive paper, we used artificial polyvinyl chloride (PVC) targets with the same dimensions as the sensitive paper labels (76 mm × 26 mm), positioned at the same height as the canopy and on the fruit clusters, to estimate the droplet deposition.

Before the applications, a polyvinyl chloride (PVC) rod was positioned on each plant, 3.6 m-long, used to fix the labels and artificial targets using a wooden nail at four different heights: top layer, middle layer, and bottom layer of the papaya canopy, and on the fruit clusters (Figure 4), respectively. Each treatment had 6 stems on 6 plants, each stem with 4 labels and 4 artificial targets, amounting to 24 sensitive paper labels and 24 PVC artificial targets per treatment.

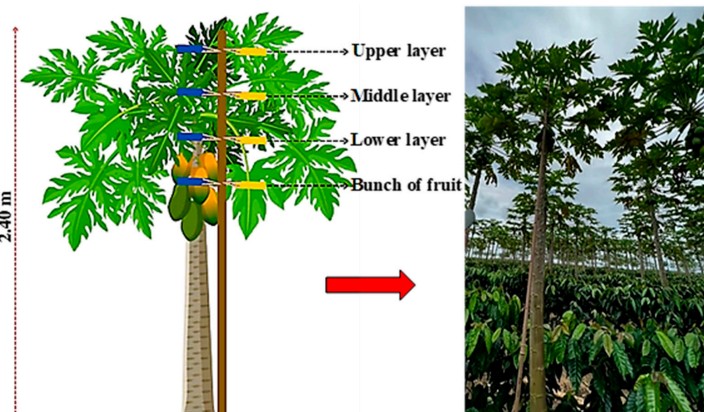

**Figure 4.** Arrangement of the sensitive paper labels (yellow labels) and artificial PVC targets (blue targets) at the respective evaluation heights using PVC rods. The red arrow indicates the manner in which the samples were collected.

The solutions sprayed in all treatments were composed of water mixed with brilliant blue pigment at a dose of 400 g ha$^{-1}$ and a non-silicone adjuvant based on balanced polymers specific for aerial applications with a low volume of syrup (0.3% v v$^{-1}$), in order to characterize the droplet spectrum on sensitive papers.

After the application of each treatment, the sensitive papers were placed in kraft paper bags, previously identified, in order to determine and quantify the impacts on labels on the same day. The PVC artificial targets were removed 30 min after the application of each treatment to ensure the solution evaporation and to leave only the pigment. After that, the artificial targets were identified and packed in plastic bags and stored in a closed Styrofoam box to avoid exposure to solar radiation and possible potential degradation by oxidation of the coloring marker.

The analyses were performed in the Mechanization and Agricultural Defensive Products Laboratory (LMDA) of the Centro Universitário Norte do Espírito Santo, of the Universidade Federal do Espírito Santo, in São Mateus, ES, Brazil.

To analyze and collect data on the droplet spectrum parameters, the hydrosensitive paper labels were scanned in a DropScope® wireless system (Figure 5). This system is made up of application programs and a wireless digital microscope with a digital image sensor with more than 2500 dpi, allowing to estimate partially overlapping drops of approximately 35 μm. The following parameters were evaluated: droplet coverage (%) and droplet density (droplets cm$^{-2}$).

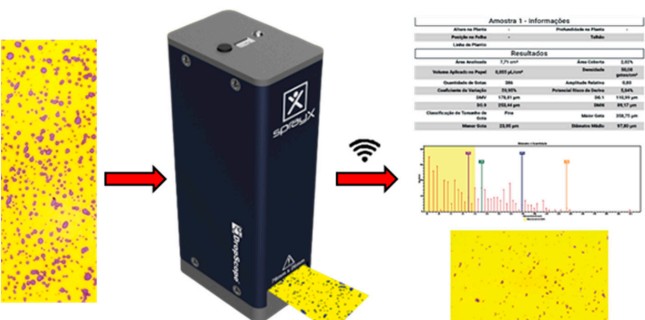

**Figure 5.** DropScope® wireless system. The red arrow indicates the manner in which the samples were collected.

In the laboratory, the artificial targets were washed to remove the tracer dye using 50 mL of distilled water per sample, added inside the plastic bags, and shaken manually for 30 s.

Absorbance readings of these solutions were taken in a spectrophotometer (Genesys 10 UV; Thermo Scientific®, Waltham, MA, USA) adjusted to measure absorbance at a

wavelength of 630 ηm. The absorbance values were obtained by reading each sample individually in the spectrophotometer and transformed into concentration (mg L$^{-1}$) using a standard curve equation developed by dilutions of 1/100, 1/200, 1/500, 1/1000, 1/2000, 1/5000, and 1/10,000 of the sample solution collected in the mixing tank before application. The mass balance generated by deposits of tracer dyes on the samples was used to estimate the deposition on artificial targets. Considering the spectrophotometer readings, the calibration curve data, and the area of the artificial targets, the amount of droplet deposition per unit area was calculated in μL cm$^{-2}$ (Equation (1)).

$$\beta_{dep} = \frac{\left(\rho_{sample} - \rho_{white}\right) \times F_{calibrate} \times V_{deposit}}{\rho_{spray} \times A_{leaf}} \tag{1}$$

where: $\beta_{deposit}$ is the deposition on artificial targets, μg cm$^{-2}$, $\rho_{sample}$ is the reading of the sample by the spectrophotometer, $\rho_{white}$ is the reading of the blank test in the spectrophotometer, $F_{calibrate}$ is the p factor of calibration, μg L$^{-1}$, $V_{deposit}$ is the liquid volume of the dilution, L, $\rho_{spray}$ is the sprayed concentration, g L$^{-1}$, and $A_{leaf}$ is the artificial target area.

Deposition data were normalized for comparison of treatments due to varying atmospheric conditions at the time of applications [35,36]. Normalized deposition was estimated according to Equation (2) [37]:

$$\beta_N = \frac{\beta_{dep} \times 10^5}{Q \times \rho_{pulveriza}} \tag{2}$$

where: $\beta_N$ is the normalized deposition (μg cm$^{-2}$ per mg ha$^{-1}$), $\beta_{dep}$ is the deposition on artificial targets (μg cm$^{-2}$), $Q$ is the application rate (L ha$^{-1}$), and $\rho_{spray}$ is the sprayed concentration (mg L$^{-1}$).

### 2.5. Monitoring Weather Conditions

The application was carried out during the morning period and the climatic conditions were monitored during the experiment using a Thal-300 Digital anemometer and an AKROM Thermo-hydro-anemometer, model KR825. The temperature varied between 24.4 °C and 29.8 °C, average relative humidity was above 60%, and wind velocity was between 2.5 and 5.6 km h$^{-1}$ at the time of application, which were adequate for spraying.

### 2.6. Statistical Analysis

The Shapiro–Wilk and Kolmogorov–Smirnov tests were performed to evaluate the regularity of the residuals. The parameters were submitted to analysis of variance. In order to compare averages when factors or interactions were significant among treatments, Tukey's test was used. All analyses were performed using the software R CORE 2022 [38] considering a 5% significance level.

## 3. Results

The results will be presented using the analysis of variance table (ANOVA), normality tests, and a figure indicating the interaction (layers and bunches of fruit XR110015 nozzle layers and bunches of fruit MGA015 nozzle) between the spray nozzles and application rates, for each parameter evaluated.

### 3.1. Drop Coverage

There was a significant interaction between the spray nozzles and application rates in the different layers and in the fruit clusters (Table 3), and the application rates were significant at $p < 0.001$ in all collection positions. The spray nozzle factor on the fruit clusters was not significant, however, there was an interaction at $p < 0.05$ between the application rates.

**Table 3.** Normality and analysis of variance (ANOVA) for droplet coverage at different plant layers and fruit bunches of papaya.

| | | | | | | | | | |
|---|---|---|---|---|---|---|---|---|---|
| **Normality of Variances [a]** | | | | | | | | | |
| **Testing** | | **Top Layer** | | **Middle Layer** | | **Lower Layer** | | **Fruit Bunches** | |
| S | | 0.303 | | 0.473 | | 0.411 | | 0.941 | |
| C | | −1.238 | | −1.061 | | −1.145 | | 0.470 | |
| W | | 0.931 [ns] | | 0.921 [ns] | | 0.934 [ns] | | 0.930 [ns] | |
| D | | 0.147 [ns] | | 0.156 [ns] | | 0.132 [ns] | | 0.132 [ns] | |
| **ANOVA [b]** | | | | | | | | | |
| Factor | GL | *p*-value | CV% | *p*-value | CV% | *p*-value | CV% | *p*-value | CV% |
| P | 1 | <0.001 *** | 23.50 | <0.001 *** | 21.23 | <0.001 *** | 14.07 | 0.072 [ns] | 42.12 |
| T | 2 | <0.001 *** | 4.04 | <0.001 *** | 3.69 | <0.001 *** | 2.11 | <0.001 *** | 5.34 |
| P × T | 2 | <0.001 *** | 1.71 | <0.001 *** | 2.36 | <0.001 *** | 1.61 | <0.05 * | 6.26 |

[a] Shapiro–Wilk test (W), Kolmogorov–Smirnov test (D), test for normality by kurtosis (C), test for normality by symmetry (S). ANOVA factors: Spray nozzles (P), application rate (T), interaction between spray nozzle and application rate (P x T). [b] * Significant at the $p < 0.05$ level, *** significant at the $p < 0.001$ level, non-normally distributed residuals, non-homogeneous variances, non-independent residuals, and rejection of the H0 hypothesis, all at the significance level. [ns] Not significant, normally distributed residuals, homogeneous variances, independent residuals, and acceptance of the H0 hypothesis, all at the 0.05 significance level. CV (%): coefficient of variance of the factors by ANOVA.

The symmetry values ranged from 0.303 to 0.473 and the kurtosis values ranged from −1.061 to 0.470. These values indicate that the droplet coverage in the respective collection positions presents normal distribution, compared to their respective averages. The data normality was confirmed by the Shapiro–Wilk and Kolmogorov–Smirnov tests, with probabilities between 0.10 and 0.50, so the hypothesis of error normality was not rejected at the 5% significance level. The variation coefficients (CV%) of the ANOVA for each factor indicate the precision of the planning and obtaining the experimental data.

For droplet coverage on papaya plants, a significant difference was observed among the application rates, 12.0, 15.0, and 18.0 L ha⁻¹, distributed by the XR110015 and MGA015 nozzles (Figure 6).

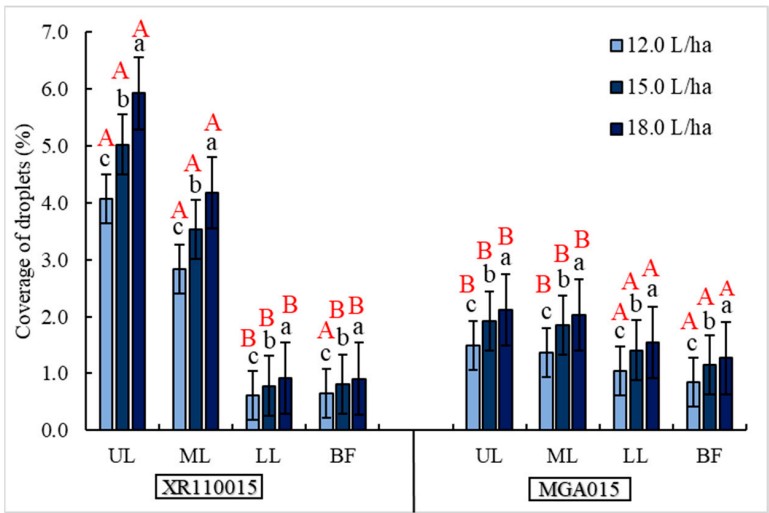

**Figure 6.** Droplet coverage (%) in the different collection positions: upper layer (UL), middle layer (ML), lower layer (LL), and bunch of fruit (BF) of papaya. Distinct letters: lower case differ among the application rates and upper case differ among the spray nozzles in the respective collection layers, Tukey's test at $p \leq 0.05$.

In the top, middle, bottom, and fruit clusters, the 18 L ha⁻¹ application rate showed the highest droplet coverage, followed by 15.0 and 12.0 L ha⁻¹, in both spray nozzles. The

XR110015 nozzle had the highest droplet coverage in the upper and middle layers, and the lowest coverage in the lower layer and on the fruit. Considering the upper canopy layer in isolation between the spray nozzles (UL XR × UL MGA), when the XR110015 nozzle was used, there was more coverage, i.e., the differences resulted in a significant increase, equivalent to 64.40% (18 L ha$^{-1}$), 62.0% (15 L ha$^{-1}$), and 63.41% (12 L ha$^{-1}$), compared to the MGA015 nozzle.

Analyzing the droplet coverage in the middle layer of the papaya plant (ML XR × ML MGA), the same trend was followed as in the upper layer, where the XR110015 nozzle had a higher droplet coverage compared to the MGA015 nozzle. Significant differences were found when applying the 18 L ha$^{-1}$ rate, with an increase using XR110015 nozzles of 52.38%, 45.71% (15 L ha$^{-1}$), and 50.0% (12.0 L ha$^{-1}$), compared to MGA015 nozzles.

Conversely to XR110015, MGA015 showed less droplet coverage in the upper and middle layers, but more coverage in the lower layer and on papaya fruit clusters. The increase in droplet coverage in the lower layer when 18 L ha$^{-1}$ was applied was 43.75%, and 42.85% and 40.0% at rates of 15.0 and 12.0 L ha$^{-1}$, respectively, compared to XR110015.

Analyzing the droplet coverage on papaya fruit clusters between spray nozzles (BF XR × BF MGA) showed the same pattern as the lower layer. Using the MGA015 nozzle at 18 L ha$^{-1}$, there was a 23.34% increase, and when applying 15 L ha$^{-1}$, the increase was 29.56% compared to the XR110015 nozzle. However, the MGA015 nozzle had the highest droplet coverage (0.85%), i.e., a 23.53% increase compared to the XR110015 nozzle.

### 3.2. Droplet Deposition

There were differences in droplet deposition among the spray nozzle and application rate factors in all collection positions in papaya (Table 4). The symmetry and kurtosis values indicated normal distribution among the analyzed means. This was confirmed by the Shapiro–Wilk normality test with values ranging from 0.813 to 0.971 and the Kolmogorov–Smirnov test with a range of 0.079 to 0.202.

**Table 4.** Normality and analysis of variance (ANOVA) for droplet deposition in different layers of plants and fruit bunches of papaya.

| Normality of Variances [a] | | | | |
|---|---|---|---|---|
| **Testing** | **Top Layer** | **Middle Layer** | **Lower Layer** | **Fruit Bunches** |
| S | 1.516 | 0.213 | 0.956 | 0.031 |
| C | 1.992 | −0.982 | −0.030 | −1.223 |
| W | 0.813 [ns] | 0.971 [ns] | 0.878 [ns] | 0.958 [ns] |
| D | 0.202 [ns] | 0.079 [ns] | 0.171 [ns] | 0.116 [ns] |

| ANOVA [b] | | | | | | | | | |
|---|---|---|---|---|---|---|---|---|---|
| Factor | GL | *p*-value | CV% | *p*-value | CV% | *p*-value | CV% | *p*-value | CV% |
| P | 1 | <0.01 ** | 48.00 | <0.01 ** | 28.15 | <0.01 ** | 53.83 | <0.01 ** | 29.50 |
| T | 2 | <0.01 ** | 26.93 | <0.001 *** | 4.07 | <0.001 *** | 5.80 | <0.001 *** | 8.52 |
| P × T | 2 | <0.05 * | 24.53 | <0.001 *** | 3.19 | <0.001 *** | 6.21 | <0.05 * | 8.35 |

[a] Shapiro–Wilk test (W), Kolmogorov–Smirnov test (D), test for normality by kurtosis (C), test for normality by symmetry (S). ANOVA factors: Spray nozzles (P), application rate (T), interaction between spray nozzle and application rate (P × T). [b] * Significant at the *p* < 0.05 level, ** significant at the *p* < 0.01 level, *** significant at the *p* < 0.001 level, non-normally distributed residuals, non-homogeneous variances, non-independent residuals, and rejection of the H0 hypothesis, all at the significance level. [ns] Not significant, normally distributed residuals, homogeneous variances, independent residuals, and acceptance of the H0 hypothesis, all at the 0.05 significance level. CV (%): coefficient of variance of the factors by ANOVA.

In the same pattern as the droplet coverage, the XR110015 nozzle showed higher droplet deposition in the upper and middle layers, statistically different from the MGA015 nozzle, which had higher droplet deposition in the lower layer and on the papaya fruit clusters (Figure 7).

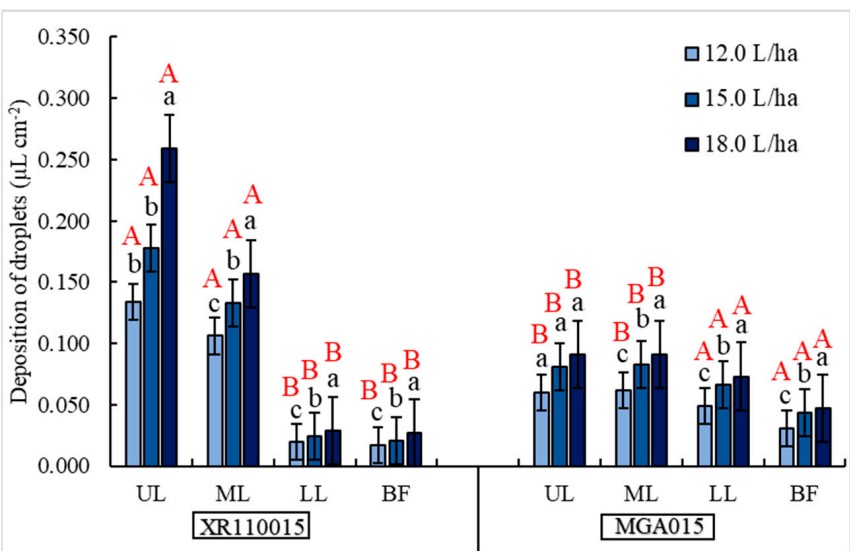

**Figure 7.** Droplet deposition ($\mu$L cm$^{-2}$) in the different collection positions: upper layer (UL), middle layer (ML), lower layer (LL), and bunch of fruit (BF) of the papaya tree. Distinct letters: lower case differ between the application rates and upper case differ between the spray nozzles in the respective collection layers, Tukey's test at $p \leq 0.05$.

When using the XR110015 nozzle, the maximum deposition was found on the upper layer of the papaya plants. The difference between the highest deposition on the upper layer (0.259 $\mu$L cm$^{-2}$) and the lowest deposition on the fruit bunches (0.027 $\mu$L cm$^{-2}$) was 0.232 $\mu$L cm$^{-2}$, equivalent to 89.58%, when applied at a rate of 18.0 L ha$^{-1}$, which had the highest deposition in all collection positions.

For the MGA015 nozzle, the highest deposition was also found in the upper layer (0.091 $\mu$L cm$^{-2}$), when applied at a rate of 18.0 L ha$^{-1}$. Using the same application rate, the lowest droplet deposition was on the fruit clusters (0.047 $\mu$L cm$^{-2}$), showing a significant difference of 0.044 $\mu$L cm$^{-2}$, equivalent to 48.35%.

When separately analyzing the lower layer of the papaya plants between the spray nozzles (LL XR × LL MGA), the MGA015 nozzle showed significantly higher droplet deposits: 0.073 (18.0 L ha$^{-1}$), 0.066 (15.0 L ha$^{-1}$), and 0.049 $\mu$L cm$^{-2}$ (12.0 L ha$^{-1}$). That is, 2.52, 2.75, and 2.45 times higher at application rates of 18.0, 15.0, and 12.0 L ha$^{-1}$, respectively, compared to the XR110015 nozzle, that showed lower values: 0.029 (18.0 L ha$^{-1}$), 0.024 (15.0 L ha$^{-1}$), and 0.020 $\mu$L cm$^{-2}$ (12.0 L ha$^{-1}$).

Isolated analysis of droplet deposition on papaya fruit clusters (BF XR × BF MGA) showed the highest deposition for the MGA015 nozzle: 0.047 (18.0 L ha$^{-1}$), 0.043 (15.0 L ha$^{-1}$), and 0.031 $\mu$L cm$^{-2}$ (12.0 L ha$^{-1}$), differing statistically from the XR110015 nozzle, which showed lower values of 0.027 (18.0 L ha$^{-1}$), 0.021 (15.0 L ha$^{-1}$), and 0.017 $\mu$L cm$^{-2}$ (12.0 L ha$^{-1}$).

*3.3. Droplet Density*

In the droplet density variable, the analyzed factors had a direct interaction in the different collection positions, showing a significant interaction ($p < 0.001$) between the factors spray nozzle and application rates (Table 5). The values of symmetry ranged from 0.033 to 0.387 and kurtosis ranged from $-1.034$ to $-1.577$, indicating normal distribution of the data in relation to the means. The coefficients of variation among the interactions were <10%, indicating high precision of the experiment performed.

**Table 5.** Normality and analysis of variance (ANOVA) for droplet density at different plant layers and fruit clusters of papaya.

| Normality of Variances [a] | | | | | | | | |
|---|---|---|---|---|---|---|---|---|
| Testing | | Top Layer | | Middle Layer | | Lower Layer | | Fruit Bunches |
| S | | 0.350 | | 0.387 | | 0.033 | | 0.295 |
| C | | −1.577 | | −1.358 | | −1.034 | | −1.454 |
| W | | 0.852 [ns] | | 0.894 [ns] | | 0.975 [ns] | | 0.909 [ns] |
| D | | 0.214 [ns] | | 0.217 [ns] | | 0.078 [ns] | | 0.169 [ns] |
| ANOVA [b] | | | | | | | | |
| Factor | GL | *p*-value | CV% | *p*-value | CV% | *p*-value | CV% | *p*-value | CV% |
| P | 1 | <0.001 *** | 45.88 | <0.001 *** | 37.10 | <0.01 ** | 27.35 | <0.001 *** | 15.47 |
| T | 2 | <0.001 *** | 6.82 | <0.001 *** | 2.58 | <0.001 *** | 5.68 | <0.001 *** | 1.80 |
| P × T | 2 | <0.001 *** | 6.20 | <0.001 *** | 4.29 | <0.001 *** | 2.88 | <0.001 *** | 2.14 |

[a] Shapiro–Wilk test (W), Kolmogorov–Smirnov test (D), test for normality by kurtosis (C), test for normality by symmetry (S). ANOVA factors: Spray nozzles (P), application rate (T), interaction between spray nozzle and application rate (P × T). [b] ** significant at the $p < 0.01$ level, *** significant at the $p < 0.001$ level, non-normally distributed residuals, non-homogeneous variances, non-independent residuals, and rejection of the H0 hypothesis, all at the significance level. [ns] Not significant, normally distributed residuals, homogeneous variances, independent residuals, and acceptance of the H0 hypothesis, all at the 0.05 significance level. CV (%): coefficient of variance of the factors by ANOVA.

The values of the Shapiro–Wilk and Kolmogorov–Smirnov tests show the normality of the data analyzed. Thus, as in the coverage and droplet density, the normality results of the data are important assumptions for the analysis of variance (ANOVA).

In droplet density, corroborating with coverage and droplet deposition, the XR110015 nozzle showed higher droplet density in the upper and middle layers, differing statistically from the MGA015 nozzle that showed higher droplet deposition in the lower layer and on the fruit clusters (Figure 8).

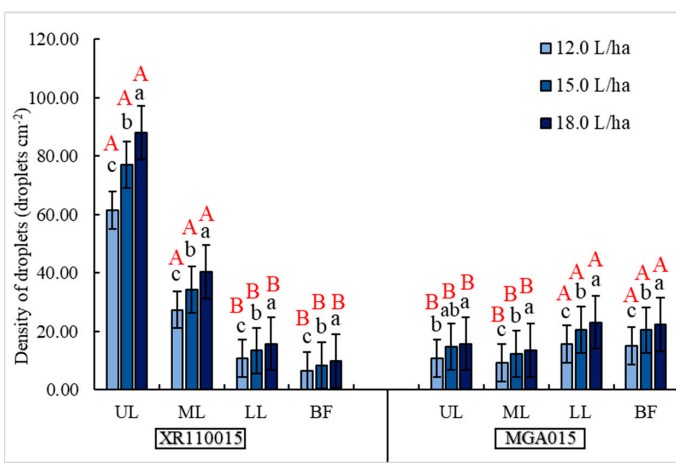

**Figure 8.** Droplet density (droplets cm$^{-2}$) in the different collection positions: upper layer (UL), middle layer (ML), lower layer (LL), and bunch of fruit (BF) of papaya. Distinct letters: lower case differ among the application rates and upper case differ among the spray nozzles in the respective collection layers, Tukey's test at $p \leq 0.05$.

Using the nozzle XR110015 at 18 L ha$^{-1}$, the highest density of drops was found in the upper layer (87.93 drops cm$^{-2}$) and the lowest in the fruit clusters (9.84 drops cm$^{-2}$), a difference of 78.09 drops cm$^{-2}$, equivalent to 88.80% higher. Moreover, at the same application rate, however using the MGA015 nozzle in the upper layer, the density of drops was lower (15.78 drops cm$^{-2}$), the difference was significant, and the XR110015 nozzle provided about 5.57 times more drops/cm$^{-2}$, i.e., 82.05% greater. However, when 15.0 L

ha$^{-1}$ was applied, the amount of drops was statistically equal (14.64 drops cm$^{-2}$) to the application rate of 18 L ha$^{-1}$, provided by the empty cone jet nozzle.

Analyzing the droplet density in the lower layer (LL XR × LLL MGA) and on the fruit clusters (BF XR × BF MGA) of the papaya plants alone, the MGA015 nozzle showed a higher droplet density in the lower layer compared to the XR110015 nozzle. The differences were significant between application rates and spray nozzles. Using the MGA015 nozzle, the highest droplet densities were found in the lower layer: 23.10 (18.0 L ha$^{-1}$), 20.50 (15.0 L ha$^{-1}$), and 15.60 droplets cm$^{-2}$ (12.0 L ha$^{-1}$), and on the fruit clusters: 22.46 (18.0 L ha$^{-1}$), 20.42 (15.0 L ha$^{-1}$), and 15.11 drops cm$^{-2}$ (12.0 L ha$^{-1}$), and higher values compared to the XR110015 nozzle at the same collection positions: in the lower layer the values were 15.80 (18.0 L ha$^{-1}$), 13.40 (15.0 L ha$^{-1}$), and 10.70 drops cm$^{-2}$ (12.0 L ha$^{-1}$), and in the fruit clusters they were 9.84 (18.0 L ha$^{-1}$), 8.28 (15.0 L ha$^{-1}$), and 6.51 drops cm$^{-2}$ (12.0 L ha$^{-1}$).

The differences in the amount of droplets/cm$^{-2}$ using the MGA015 nozzle in the lower layer were equivalent to 31.60% (18.0 L ha$^{-1}$), 34.63% (15.0 L ha$^{-1}$), and 31.41% (12.0 L ha$^{-1}$), and in the fruit bunches were equivalent to 56.19% (18.0 L ha$^{-1}$), 59.45% (15.0 L ha$^{-1}$), and 56.91% (12.0 L ha$^{-1}$), compared to the XR110015 nozzle in both collection positions.

### 3.4. Volumetric Median Diameter

In the volumetric median diameter (VMD), the interactions between spray nozzles and application rates were significant in the upper layer, lower layer, and in the fruit clusters (Table 6). However, in the middle layer, the interaction between nozzle × rate was not significant, indicating homogeneity of variances and acceptance of the hypothesis of equality among treatments.

**Table 6.** Normality and analysis of variance (ANOVA) for volumetric median diameter at different plant layers and clusters of papaya fruit.

| Normality of Variances [a] | | | | | | | |
|---|---|---|---|---|---|---|---|
| Testing | | Top Layer | | Middle Layer | | Lower Layer | | Fruit Bunches |
| S | | 0.427 | | 0.269 | | 0.526 | | −0.009 |
| C | | −0.514 | | −0.802 | | −0.299 | | −0.870 |
| W | | 0.969 [ns] | | 0.973 [ns] | | 0.963 [ns] | | 0.982 [ns] |
| D | | 0.103 [ns] | | 0.083 [ns] | | 0.116 [ns] | | 0.085 [ns] |
| ANOVA [b] | | | | | | | |
| Factor | GL | *p*-value | CV% | *p*-value | CV% | *p*-value | CV% | *p*-value | CV% |
| P | 1 | <0.01 ** | 12.36 | <0.05 * | 11.59 | <0.05 * | 19.35 | 0.298 [ns] | 29.70 |
| T | 2 | <0.001 *** | 6.95 | <0.001 *** | 3.11 | <0.001 *** | 6.47 | <0.001 *** | 3.40 |
| P × T | 2 | <0.05 * | 5.22 | 0.121 [ns] | 3.63 | <0.05 * | 4.40 | <0.05 * | 3.02 |

[a] Shapiro–Wilk test (W), Kolmogorov–Smirnov test (D), test for normality by kurtosis (C), test for normality by symmetry (S). ANOVA factors: Spray nozzles (P), application rate (T), interaction between spray nozzle and application rate (P × T). [b] * Significant at the $p < 0.05$ level, ** significant at the $p < 0.01$ level, *** significant at the $p < 0.001$ level, non-normally distributed residuals, non-homogeneous variances, non-independent residuals, and rejection of the H0 hypothesis, all at the significance level. [ns] Not significant, normally distributed residuals, homogeneous variances, independent residuals, and acceptance of the H0 hypothesis, all at the 0.05 significance level. CV (%): coefficient of variance of the factors by ANOVA.

In the fruit clusters, the spraying nozzle factor was not significant, however, the application rate factor was significant at $p < 0.001$, which explains the reason for the interaction between the factors ($p < 0.05$) in this collection position.

Overall, the VMD values were directly proportional to the application rates (18.0 > 15.0 > 12.0 L ha), i.e., the higher the application rate, the higher the VMD average values in the different sampling positions (Table 7).

**Table 7.** Normality and analysis of variance (ANOVA) for the volumetric median diameter in the different layers of the plant and clusters of papaya fruits.

| Position | Nozzle | Application Rate L/ha | | |
| --- | --- | --- | --- | --- |
| | | 12.0 | 15.0 | 18.0 |
| UL | XR110015 | 190.76 ± 27.53 bC | 238.45 ± 34.42 bB | 298.01 ± 42.04 bA |
| | MGA015 | 240.51 ± 39.42 BC | 325.02 ± 52.28 aB | 357.52 ± 58.60 yr |
| ML | XR110015 | 224.25 ± 41.76 ns | 284.33 ± 50.41 ns | 326.93 ± 42.64 ns |
| | MGA015 | 187.42 ± 31.07 ns | 253.28 ± 41.99 ns | 278.61 ± 46.19 ns |
| LL | XR110015 | 184.88 ± 30.71 aC | 224.82 ± 47.50 aB | 265.29 ± 56.05 aA |
| | MGA015 | 142.37 ± 26.16 bC | 192.40 ± 35.35 aB | 211.64 ± 38.89 bA |
| BF | XR110015 | 215.15 ± 22.53 BC | 272.24 ± 22.82 aB | 321.24 ± 26.93 aA |
| | MGA015 | 187.84 ± 43.19 aC | 253.83 ± 58.36 aB | 279.22 ± 64.20 aA |

Averages followed by different letters, lower case in the columns and upper case in the rows, significantly differ from each other by Tukey's test, $p \leq 0.05$; ns, not significant. Upper (UL), middle (ML), and lower (LL) layers, and on papaya fruit clusters (BF).

In the upper layer, the MGA015 nozzle presented higher values in droplet diameter, with a higher VMD (357.53 μm), compared to the XR110015 nozzle, using 18 L ha. In the middle layer, the results were not significant between the spray nozzles and application rates, as the ANOVA showed; however, the XR110015 nozzle showed higher VMD values compared to the MGA015 nozzle. In the lower layer, it follows the same pattern, the XR110015 nozzle had higher values compared to MGA015 when applying 12.0 and 18.0 L ha, however, when applying 15.0 L ha, the droplet diameter was statistically equal. Finally, in the fruit clusters, although the spray nozzle factor was statistically equal (not significant), the mean VMD values were different and increased proportionally as the application rates increased.

## 4. Discussion

The results presented indicate that the application rate had a significant effect on the droplet distribution along the layers and fruit clusters of the papaya plants. Regardless of the target collection position, the coverage, density, and deposition of drops, as well as the volumetric median diameter (VMD), increased proportionally as the application rates increased. Similar results were found in other crops, such as in wheat [39], cotton [40], sugarcane [41], and Nanguo pear orchards [20].

Although the application rate of 18 L ha$^{-1}$ provided higher values in the variables analyzed, the application rate of 15 L ha$^{-1}$ showed considerable values along the canopy and in the fruit clusters, considering that at 12 L ha$^{-1}$, the droplet distribution was mainly limited in the upper layer, indicating that the lower application rate can be used for applications where the target is located with higher incidence in the upper layer of the papaya tree, i.e., in young leaves, as in the control of the white mite (*Polyphagotarsonemus latus*). However, in the papaya culture, the mites are the main plague that are mostly located on the abaxial side of the leaf. In the present research, the distribution of drops on the leaf was not evaluated.

That said, in the applications using UAVs, one seeks to decrease application rates while increasing the operational capacity of the equipment per unit area, considering that the equipment is limited by battery life and load capacity: the initial payload range is 8–15 L, and some current models can reach 20–40 L [42,43]. However, depending on the incidence pressure and location of the pest and/or disease on the crop plant, lower application rates will not always have better efficacy, as considered in [44] on wheat plants and recently in [45] on Arabica coffee plants; although, intermediate application rates are promising, as observed by the aforementioned authors.

This study showed significant differences between the spray nozzles in the variables analyzed, as well as interactions between the application rates. For example, in the coverage, density, and deposition of drops, the XR110015 nozzle had higher values in the upper and middle layers and lower values in the lower layer and fruit clusters. On the other hand, the MGA015 nozzle displayed inverse values, with lower values in the upper and middle layers and higher values in the lower layer and on the fruit clusters.

The flat-fan spray nozzles distribute the liquid transversely, practically symmetrical in relation to the center of the nozzle; due to this, with the opening of the fan, it was observed during the applications that the drops were concentrated in the first layers of the papaya plants. When using the XR110015 nozzle at 18 L ha$^{-1}$, the highest droplet coverage was in the upper layer (5.9%) of the papaya plants, and there was a difference compared to the lower layer and on the fruit clusters of 6.56 times greater, equivalent to 84.74%. Similar results were found in shrub orchards as in mango [28], almond [33], and guava [27]. When the experiment was carried out, the papaya plants were in the adult phase, which resulted in a reduction in the number of leaves, and these were concentrated in a more compact group, consisting of large leaves in the apical region. This is explained by the fact that during the applications, the so-called "umbrella effect" occurred, in which the leaves in the upper and middle layers prevented the increase in coverage, density, and deposition of drops in the lower layer and fruit clusters. The authors of [46], using a UAV equipped with the XR110-01 and -015 nozzles, to control the caterpillar (*Spodoptera frugiperda*) on corn plants, observed that coverage and droplet deposition were higher in the upper layer and lower in the lower layer. Other studies support this statement using UAV-applied flat-fan nozzles, such as on citrus [47] and on pigeon pea [48].

Hollow cone nozzles, such as the MGA015, are characterized by a greater deposition of liquid on the outside of the cone, since there are practically no droplets in the center, and are widely used for applications in fruit trees. During the applications in a wind tunnel using UAV that greater turbulence in the distribution of drops was generated, due to the swirl chamber present in the hollow cone nozzles [49]. Thus, during the applications on papaya plants, a better droplet distribution was observed along the canopy and on the papaya fruit clusters.

Furthermore, the downwash effect contributed to the penetration and distribution of the drops on the plants, i.e., the air flow generated by the multi-rotors agitates the leaves on the plant canopy, facilitating the deposition process, especially in the lower layer [18,45]. Despite providing lower values in the upper and middle layers, compared to the XR110015 nozzle, the MGA015 nozzle displayed better droplet distribution in the lower layer and on the papaya fruit clusters, indicating possible control of pests and diseases located in this region.

The volumetric median diameter (VMD) had an inversely proportional relation to the analyzed variables, i.e., as the droplet size increased, less coverage, deposition, and droplet density were observed. For example, in the lower layer and fruit clusters, the MGA015 nozzle showed lower values in the droplet diameter and consequently provided greater coverage, density, and deposition of droplets in both collection positions on papaya plants. The typical feature of hollow cone nozzles provides droplets with smaller diameters and enhances the uniformity of the application [50]. Similar results were found in [51], using a UAV for applications on rice plants, and in [52] using hollow cone nozzles at different pressures in a turboatomizer sprayer on 'Formosa' papaya plants.

The VMD was directly proportional to the application rates: as the application rates increased, the VMD values increased in the different collection positions. Similar results using a UAV were found in [25] in apple orchards. As an example, at the application rate of 45 L ha (lowest rate), the VMD was 167.4 μm, and at 100 L ha (highest rate) the VMD was 292.2 μm. Overall, in the different collection positions, when using the XR11015 nozzles, the droplet diameter ranged from 184.88 to 326.93 μm, the MGA015 nozzle provided drops with a volumetric median diameter of 142.37 to 357 μm, and both nozzles provided from fine to medium drops [53,54] when these were already on target, being subject to primary and secondary drift in inappropriate weather conditions. The values described are in accordance with the droplet classification described by the nozzles' manufacturers.

Although the results of this study were significant for the use of the UAV in the distribution of drops on papaya plants, the morphological architecture of papaya plants is different according to the stages of plant development. However, for applications using UAVs, the canopy structure of the crop plant directly impacts the settings of the

operational parameters, as considered in [31,55,56] and recently in [26]. In this regard, further experimental studies are needed to elucidate new operational parameters, such as new application rates, operational flight speed and height, spray nozzles, and flight paths, in order to increase the application efficiency and effectiveness in pest and disease control in papaya crops.

## 5. Conclusions

The use of unmanned aerial vehicles has been recognized as an important tool in spraying and applications in orchards due to a series of advantages, especially the applicability in adverse conditions when the access of tractor-drawn equipment is not possible, the greater efficiency of droplet deposition and the effectiveness of the products applied, as well as being sustainable. Therefore, the main purpose of this study was to evaluate the effect of spray nozzles and application rates on droplet distribution along the canopy height and fruit clusters of papaya plants by means of a UAV. After the methodological application, data collection, and analysis of the results obtained, we can conclude that:

(1) Regardless of the application rate and spray nozzle used, the droplet distribution along the layers and on the clusters of papaya fruits provided by the UAV was uniform, especially in the upper leaf layer.

(2) The MGA015 hollow cone spray nozzle at 15.0 and 18.0 L ha$^{-1}$ can be used as a reference for future evaluations of application efficiency with fungicides and/or insecticides, because it showed better droplet distribution at 18 L ha$^{-1}$ in the lower layer and on the fruit clusters.

(3) The XR110015 nozzle had better droplet distribution in the upper and middle canopy layers at the application rate of 12 L ha$^{-1}$, and the control on targets located in the first layers of the papaya canopy can be effective.

(4) Considering the deposition of drops in orchards of different fruit trees, one can infer that the use of UAVs in the papaya crops may be feasible; however, further studies should be conducted in order to test other hollow cone spray nozzles, carry out comparisons with other forms of application, and analyze the control of pests and/or diseases, because there is a tendency to improve the distribution of drops along the canopy of papaya using conical tips. In addition, different stages of development of the papaya plant should be considered.

**Author Contributions:** Conceptualization, E.L.d.V. and L.F.O.R.; methodology, E.L.d.V., L.F.O.R. and G.G.S.J.; software, L.F.O.R.; validation, E.L.d.V. and L.F.O.R.; formal analysis, L.F.O.R.; investigation, E.L.d.V. and G.G.S.J.; resources, E.L.d.V., L.F.O.R. and G.G.S.J.; data curation, L.F.O.R.; writing—original draft preparation, L.F.O.R. and E.L.d.V.; writing—review and editing, E.L.d.V., P.C. and Y.L.; visualization, E.L.d.V., P.C., Y.L. and L.F.O.R.; supervision, E.L.d.V.; project administration, E.L.d.V.; funding acquisition, E.L.d.V. All authors have read and agreed to the published version of the manuscript.

**Funding:** This study was financed in part by the Coordenação de Aperfeiçoamento de Pessoal de Nível Superior—Brazil (CAPES 18/2020), and Fundação de Amparo a Pesquisa do Espírito Santo (FAPES 140/2021)—Brazil. CAPES/FAPES Cooperation—Postgraduate Development Program—PDPG.

**Data Availability Statement:** Not applicable.

**Conflicts of Interest:** The authors declare no conflict of interest.

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
