# Peer review of "Impact of Operational Parameters on Droplet Distribution Using an Unmanned Aerial Vehicle in a Papaya Orchard"

_agronomy, doi:10.3390/agronomy13041138_

Round 1

Reviewer 1 Report

The article entitled “Impact of operational parameters on droplet distribution using an unmanned aerial vehicle in a papaya orchard” illustrates the use of drones to distribute plant protection products to prevent disease in papaya plantations in the State of Santo Spirito in Brazil.

Using drones is undoubtedly a modern alternative to traditional agricultural practices. Other studies cited by the authors refer to the use of these technologies on various fruit crops (mango, citrus, pineapple, pear, apple, etc.).

This study intends to evaluate the effectiveness of two spray nozzles applied on the UAV with different rates of application of plant protection products (12, 15, 18 L/Ha). The variables droplet coverage, droplet density, droplet distribution and droplet diameter along the trees were measured using a digital microscope to analyze the droplet spectrum parameters mentioned above. Statistical tests validated the results.

The manuscript presents an extensive state of the art; all sections are well developed. Check the English language because some errors need to be corrected. After these minor revisions, the paper can be accepted.

in detail some corrections to be made:

Keywords: the following keywords, uniformity and efficiency) are not representative of the subject of the paper.

Line 50: in my opinion delete by

Line 71: please add “a”significant…..

Line 74-76: Please change……. In vineyards, [29] observed that high speed (3 m s-1) enabled increased droplet deposition on the canopy and reduced losses to non-target areas using conventional spray nozzles. Furthermore…..

Line 91 Change in Regarding UAVs………….

Line 92 please change in…. application quality

Line 92: Change in This study aimed to evaluate………….

Line 123 treatment

Line 135-137: Please change in Water-sensitive paper labels with 76 × 26 mm dimensions were used to determine the sprayed droplet spectrum and to estimate the application efficiency variables: coverage, droplet density and the volumetric median diameter, respectively.

Line 182-184: Please rewrite this sentence

Line 203-204 rewrite in….. These parameters were submitted to the variance investigation to analyze the results obtained.

Line 224. Please delete the planning

Line 256 Please change in ha−1

Line 288 Please change in ha−1

Sometimes you write 15 L ha-1, other times 15 L/ha. Please standardize your writing.

Line 385. Please change in: However, the XR110015 nozzle showed higher VMD values than the MGA015 nozzle.

Line 386-388: I suggest to change in this way: In the lower layer, it follows the same pattern. The XR110015 nozzle had higher values than MGA015 when applying 12.0 and 18.0 L ha. However, when applying 15.0 L ha, the droplet diameter was statistically equal.

Line 407 Please add “a”… with a higher

Line 425 layers

Line 426 Please change distributes in distribute

Line 427 Please delete in relation

Line 448-450: I suggest to change in this way…..Thus, a better droplet distribution was observed along the canopy and on the papaya fruit clusters during the applications on papaya plants.

Line 485 crops

Author Response

Dear,

The authors welcome your comments, observations, and suggestions. We suggest reading the answers to the specific questions presented in the appendix.

Reviewer 2 Report

The paper agronomy-2301835, “Impact of operational parameters on droplet distribution using an unmanned aerial vehicle in a papaya orchard”, resumes interesting information on the potential use of unmanned aerial vehicles for agricultural purposes.

If I correctly read the paper, the authors study the potential for using unmanned aerial vehicles in a papaya orchard for a smart agriculture approach. Overall I think this is a good attempt at studying this interesting topic about effectively improving production under an orchard setup.

However, I think some improvements could be made.

The title is not fully pertinent to the topics of the manuscript. The Abstract layout could be improved to demonstrate the purpose or objectives of the study and then present the key findings. Similarly, the introduction could be changed with some of the information moved to the body of the paper.

The review was fairly comprehensive and had a logical flow.

Notwithstanding the scientific sound of this paper, the presentation in the form of a manuscript is confusing, unclear, and requires several adjustments. Some sentences in several sections should have reference citations included.

The abstract and the conclusion are two important parts of the manuscript; I suggest rewriting them.

Use the International System of Units

Verify the correct use of punctuation marks.

Author Response

(The authors gave the same response as above.)

Reviewer 3 Report

This paper does not offer enough scientific progress. While the study provides some insights into the efficiency of droplet distribution using an unmanned aerial vehicle in a papaya orchard, the findings are not sufficiently novel or significant to contribute to the scientific literature in this field. Additionally, the paper lacks sufficient detail on the methodology and statistical analysis used, and the results are presented in a descriptive manner rather than a scientific format.

Author Response

(The authors gave the same response as above.)

Reviewer 4 Report

The research method of the manuscript is not new, and many similar kinds of literature can be searched. But the authors are targeting a new applied crop, Papaya. The authors need more space to describe the current state of UAV sprayer use in the region, as well as the unique characteristics of Papaya. Here are the specific comments.

1. The literature review lacks the necessary conciseness, and there are too many divided paragraphs, which can be properly merged. UAV sprayers have been tested or applied on other fruit trees. What are the similarities and differences between this study and previous studies?

2. L137 Will the experimental treatment design be affected by the difference in crop structure in different experimental areas?

3. L158 the hydrosensitive paper? Please unify the name.

4. L180-120 Missing standard curve?

5. Table 3 Variable letters should be italicized.

6. Volumetric Median Diameter is a nozzle's property, why should it be analyzed? It seems that we can predict the outcome of experiments even without conducting them. What is the author's purpose in conducting this assessment?

7. The author analyzes VMD based on the results of WSP, which seems unconvincing. During the deposition of WSP, the droplet size may change, including the spreading, merging, and bouncing of droplets.

8. L447 Is it appropriate to divide paragraphs here?

9. The Discussion section is divided into too many paragraphs, and it's hard to read the logic.

Author Response

(The authors gave the same response as above.)

Round 2

Reviewer 3 Report

MY point remains the same, so please do not send this manuscript to me again.

This paper does not offer enough scientific progress. While the study provides some insights into the efficiency of droplet distribution using an unmanned aerial vehicle in a papaya orchard, the findings are not sufficiently novel or significant to contribute to the scientific literature in this field. Additionally, the paper lacks sufficient detail on the methodology and statistical analysis used, and the results are presented in a descriptive manner rather than a scientific format.